# Role of Checkpoint Inhibitors in the Management of Gastroesophageal Cancers

**DOI:** 10.3390/cancers15164099

**Published:** 2023-08-14

**Authors:** Frederic Karim, Adina Amin, Marie Liu, Nivetha Vishnuvardhan, Saif Amin, Raffey Shabbir, Brandon Swed, Uqba Khan

**Affiliations:** 1Internal Medicine, New York-Presbyterian Brooklyn Methodist Hospital, 506 6th Street, Brooklyn, NY 11215, USA; ada9091@nyp.org (A.A.); pqe9001@nyp.org (M.L.); saif.amin@stonybrook.edu (S.A.); raffey.shabbir@gmail.com (R.S.); 2Hematology/Oncology, New York-Presbyterian Brooklyn Methodist Hospital, 506 6th Street, Brooklyn, NY 11215, USA; niv9051@nyp.org; 3Hematology/Oncology, Weill Cornell Medicine, 515 6th Street, Brooklyn, NY 11215, USA; brs2023@med.cornell.edu (B.S.); uqk9001@med.cornell.edu (U.K.)

**Keywords:** checkpoint inhibitors, esophageal cancer, gastric cancer, immunotherapy, PD-L1, HER2, clinical trials

## Abstract

**Simple Summary:**

Gastroesophageal cancers are one of the leading causes of cancer-related mortality worldwide. Recent advances in systemic therapies have led to modest improvements in survival outcomes for these patients. Specifically, immunotherapy in the form of checkpoint inhibitors (CPIs) has transformed how we treat these malignancies. CPIs have become part of standard care in treating metastatic gastroesophageal cancers. Although initially approved in later-line settings for immunotherapy-naïve patients, CPIs such as nivolumab and pembrolizumab have now been incorporated into first-line regimens. The role of CPIs in managing advanced gastroesophageal cancers continues to evolve as novel combination strategies are being explored and predictive biomarkers are further refined. This article reviews the important clinical trials that have led to current immunotherapy approvals and highlights relevant biomarkers and ongoing clinical trials incorporating CPIs in gastroesophageal cancers.

**Abstract:**

Purpose: This article reviews the essential clinical trials that have led to these immunotherapy approvals and explores the use of predictive biomarkers, such as PD-L1 expression and MSI status, to identify patients who are most likely to benefit from immunotherapies. Methods: This case review series describe findings from different clinical trials and contribute to the evolving understanding of the role of CPIs in managing advanced gastroesophageal cancers and may lead to improved treatment options and patient outcomes. Ongoing clinical trials also hold promise for expanding treatment options and improving patient outcomes in the future. Methods: The systematic review followed the recommendations of the Preferred Reporting Items for Systematic Reviews and Meta-Analyses (PRISMA). The protocol has not been registered. A systematic literature review was conducted to identify relevant clinical trials and studies that describe the role of immune checkpoint inhibitors in managing advanced gastroesophageal cancers. Electronic database (PubMed, Clinicaltrials.gov, Society of Immunotherapy of Cancer, Aliment Pharmacology & Therapeutics, BMC cancer, Molecular Cancer Research, Nature Reviews Molecular Cell Biology, American Association for Cancer Research, Science, Nature, Cancer Discovery, Journal of the National Cancer Institute, Advanced Immunology, Oncotarget, Nature Medicine, Nature Genetics, Gut, Pathology and Oncology Research, Journal of Clinical Oncology, The New England Journal of Medicine, Gastrointestinal oncology, JAMA Oncology, Journal of Gastrointestinal Oncology, Current Oncology, Annals of Oncology, The Lancet, JCO Oncology Practice, Future Oncology, Gastric Cancer, CA: A Cancer Journal for Clinicians, American Journal of Gastroenterology, Gastroenterology, Journal of the National Cancer Institute, International Journal of Epidemiology, Helicobacter, Gastroenterology Review) were searched using a combination of relevant keywords and MESH terms. The search encompassed articles published up to 5/2023. Additionally, manual searches of reference lists of selected articles and pertinent review papers were conducted to ensure comprehensive coverage of relevant studies. Studies were included if they provided insights into clinical trials evaluating the efficacy and safety of CPIs in treating advanced gastroesophageal cancers. Relevant case reviews and trials exploring combination therapies involving CPIs were also considered. Articles discussed in the utilization of predictive biomarkers were included to assess their impact on treatment outcomes. Data from selected studies were extracted to inform the narrative review. Key findings were summarized, including clinical trial designs, patient populations, treatment regimens, response rates, progression-free survival (PFS), overall survival (OS), and adverse events. The role of predictive biomarkers, particularly PD-L1 expression and MSI status, in identifying patients likely to benefit from CPIs was critically evaluated based on study results. Ongoing clinical trials investigating novel combination strategies and exploring the broader scope of CPIs in gastroesophageal cancers were also highlighted. The collected data were synthesized to provide a comprehensive overview of the crucial clinical trials that have contributed to the approval of CPIs for advanced gastroesophageal cancers. The role of CPIs in different lines of therapy, including first-line regimens, was discussed. Furthermore, the evolving landscape of predictive biomarkers was examined, emphasizing their potential significance in optimizing patient selection for CPI therapy. Ongoing clinical trials were reviewed to underscore the continuous efforts in expanding treatment options and improving patient outcomes in the future.

## 1. Introduction

Esophageal cancer ranks seventh in incidence and sixth in mortality overall. Incidence and mortality rates are two to three-fold higher in men [1]. Geographical distribution and risk factors substantially differ between the two most common histologic subtypes of esophageal cancer–squamous cell carcinoma (SCC) and adenocarcinoma (AC). Smoking and heavy alcohol consumption are major risk factors for SCC in Western countries. In contrast, nutritional deficiencies, nitrosamine, betel quid chewing, consumption of pickled vegetables and very hot foods contribute to developing countries. Excess body weight, gastroesophageal reflux disease and Barrett’s esophagus are the key risk factors for AC. The incidence of AC is rapidly rising in high-income countries due in part to increasing risk factors and decreasing incidence of chronic H. Pylori infections, which has been inversely associated with AC [2,3,4,5,6,7,8,9,10].

Gastric cancer ranks fifth in incidence and fourth in mortality globally. Although proximal (cardia) and distal (non-cardia) gastric cancers are often grouped, these are distinct entities regarding risk factors and epidemiology. Distal gastric cancers are steadily declining in incidence due to decreasing prevalence of associated risk factors like chronic H. Pylori infection and harmful food preservation methods [1,11,12]. Approximately 36% of patients with gastric cancer are diagnosed at a late stage with distant metastatic disease, with a dismal 5-year survival rate of 6%. 

Until recently, cytotoxic combination chemotherapy has been the preferred first-line treatment strategy. With advances in molecularly directed therapy, the treatment landscapes are shifting for both SCC and AC. Following their approval in later-line settings, CPIs such as pembrolizumab and nivolumab are now approved in the first-line treatment of advanced esophageal and gastric cancers. The mechanism of action for CPIs involves blocking specific checkpoint molecules, such as PD-1 or CTLA—4. When these checkpoints are blocked, the immune system can recognize and target cancer cells more effectively, leading to tumor regression. Various biomarkers have been studied to identify the subgroups of patients most likely to benefit from immunotherapy, which remains an area of active investigation. Microsatellite instability (MSI) and mismatch repair deficiency (dMMR) status have been validated as predictive biomarkers for response to immunotherapy [13,14].

In patients with MSI-high tumors, the DNA sequence of certain genes is altered due to errors in the cell’s DNA replication machinery. This genomic alteration generally results in a favorable response to immunotherapy [15]. Similarly, the level of programmed cell death-ligand 1 (PD-L1) expression has been predictive of response to immunotherapy in some studies in upper gastrointestinal cancers. PD-L1 is a protein that may be expressed on the surface of some cancer cells and suppresses the immune response, thereby allowing the cancer to evade immune surveillance. The combined positive score (CPS), which includes PD-L1 staining in tumor cells, macrophages, and lymphocytes, has been shown to better predict immunotherapy response than the tumor proportion score (TPS), which only reflects PD-L1 expression in tumor cells. In gastrointestinal malignancies, tumor infiltrating cells in the tumor microenvironment express PD-L1 at higher levels than cancer cells. The prevalence of PD-L1 expression has been reported in up to 57% of the gastric and gastroesophageal junction (GEJ) tumors [16]. Accordingly, testing for MSI status or PD-L1 expression can help identify patients who may derive benefit from immunotherapies.

With recent positive clinical trials incorporating CPIs plus chemotherapy in advanced gastroesophageal cancers, these agents have become the standard of care in the first-line setting. The CheckMate 649 trial led to the approval of nivolumab with a platinum doublet chemotherapy regimen in advanced gastroesophageal AC. Notably, the CPS ≥ 5 subgroup and MSI-high patients derived the greatest benefit. Although there was an overall survival (OS) benefit among allcomers, this was thought to be primarily driven by benefit in the CPS ≥ 5 subgroup [17]. The ATTRACTION-4 trial conducted in Asian countries in unselected patient populations demonstrated that the combination of nivolumab and chemotherapy improved progression-free survival (PFS), but there was no OS [18]. Although the optimal CPS cutoff for the addition of nivolumab remains unclear, current National Comprehensive Cancer Network (NCCN) guidelines recommend this combination for tumors with PD-L1 CPS ≥ 5. Of note, these two trials included patients with predominantly gastric AC. CheckMate 649 included 16% GEJ and 13% esophageal AC tumors, while ATTRACTION-4 trial included only 8% of GEJ tumors, suggesting a weak recommendation for adding nivolumab in this subset [17,18]. KEYNOTE-590, which predominantly included esophageal SCC patients, approved pembrolizumab with chemotherapy for metastatic or locally advanced esophageal or GEJ tumors, irrespective of CPS. NCCN recommends this combination as category 2A only for CPS ≥ 10 [19]. The CheckMate 648 trial, which included patients with advanced esophageal SCC, subjected patients to three first-line treatment arms: nivolumab plus chemotherapy, nivolumab plus ipilimumab and chemotherapy alone. Patients with a PD-L1 expression ≥ 1% treated with immunotherapy benefitted PFS and OS compared to those treated with chemotherapy alone [20]. Furthermore, in the postoperative setting, nivolumab is the only approved immunotherapy for patients with resected esophageal or gastroesophageal junction cancer who have received neoadjuvant chemoradiation followed by complete resection and confirmed to have the residual pathological disease [21]. 

As evidenced by these clinical trial findings, the role of CPIs in advanced gastroesophageal cancers is predominantly determined by the tumor location and PD-L1 score, particularly CPS. Although there are variations in outcomes among different clinical trials involving CPIs in advanced gastroesophageal cancers, most studies consistently demonstrate greater clinical benefit in tumors with higher PD-L1 expression. While CPIs are not recommended for those with a CPS of 0, their use is still controversial in patients with AC and a CPS of 1–5. In this subset of patients, it is important to consider patient-specific factors and toxicity concerns to guide therapeutic decision-making. Ongoing clinical trials may further help us select patients who will benefit most from CPIs in the first-line setting and may provide greater upfront treatment options for patients with advanced disease, which remains an unmet need.

Her2 was the first biomarker that had clinical implications in managing gastric cancers. Up to 20 percent of gastric cancers are Her2-positive, and Her2-positive rates are higher in GEJ cancers and those with intestinal histologies [22]. Based on the ToGA trial, the addition of trastuzumab to platinum + fluoropyrimidine chemotherapy resulted in an improved OS from 11.1 months with chemotherapy alone to 13.8 months with the combination (hazard ratio (HR) 0.74; 95% confidence interval (95% CI) 0.60–0.91; *p* = 0.0046) [23]. Several other Her2-directed therapeutic strategies have also been explored, notably in the KEYNOTE-811 trial, which led to the accelerated approval of pembrolizumab in combination with trastuzumab, fluoropyrimidine-, and platinum-containing chemotherapy in the first-line setting in this biomarker-selected patient population [24]. 

Several clinical trials are ongoing to evaluate immunotherapy agents like pembrolizumab, nivolumab and ipilimumab in neoadjuvant and adjuvant settings, as well as in advanced disease alone or combination with tyrosine kinase inhibitors [25,26,27,28]. Here, we provide a comprehensive overview of the role of CPIs in different lines of therapy for gastric and esophageal cancers. We also highlight several pivotal ongoing clinical trials involving immunotherapy in these tumor types, which may further expand our treatment options and lead to improved patient outcomes and quality of life. 

## 2. Role of Checkpoint Inhibitors in the First-Line Setting

Several studies were conducted to evaluate the efficacy of CPIs as a first-line treatment for different types of advanced gastric and esophageal cancers. In gastrointestinal malignancies, CPIs primarily involve PD-1/PD-L1 and CTLA-4. 

CheckMate 649 was a global, randomized, phase 3 trial that enrolled previously untreated adult patients with non-HER2-positive gastric, gastroesophageal junction and esophageal AC. Patients were randomly assigned to nivolumab plus chemotherapy, nivolumab plus ipilimumab, or chemotherapy alone. The primary endpoints were OS or PFS with a PD-L1 CPS of five or more. One thousand five hundred eighty-one patients were assigned for treatment, with 789 (50%) patients on nivolumab plus chemotherapy and 792 (50%) patients on chemotherapy alone [17]. The median follow-up for OS was 13.1 months for the nivolumab plus chemotherapy group and 11.1 months for the chemotherapy alone group. Additionally, the median PFS for patients with PD-L1 CPS ≥ 5 was 2.8 months for the nivolumab/ipilimumab group and 6.3 months for the chemotherapy group (hazard ratio 1.42). Nivolumab plus chemotherapy resulted in significant improvements in OS (hazard ratio 0.71 [98.4% CI 0.59–0.86]; *p* < 0.0001) and PFS (HR 0.68 [98% CI 0.56–0.81]; *p* < 0.0001) versus chemotherapy alone in patients with a PD-L1 CPS of five or more (minimum follow-up 12.1 months). This trial demonstrated that nivolumab plus chemotherapy and nivolumab/ipilimumab resulted in significant improvements in both OS and PFS compared to chemotherapy alone. The treatment was well-tolerated, although grade 3–4 treatment-related adverse events were more common in the nivolumab plus chemotherapy group. 

CheckMate 648 is another global, randomized, phase 3 trial that evaluated previously untreated, unresectable advanced, recurrent, or metastatic esophageal SCC. Patients were randomly assigned to receive nivolumab plus chemotherapy, nivolumab plus ipilimumab, or chemotherapy. The primary endpoints were OS and PFS with a PD-L1 expression of 1% or greater. The secondary endpoints comprised OS and PFS in all randomized patients and objective response rate in patients with tumor cell PD-L1 expression greater than or equal to 1%. A total of 970 patients underwent randomization. The study showed that the median OS in the nivolumab/chemotherapy group was 12.8 months (95% CI, 11.1–15.7) compared with 10.7 months (95% CI, 9.5–12.1) for chemotherapy alone. For patients with a PD-L1 expression of 1% or higher, the median OS again favored the nivolumab group at 15.0 vs 9.1 months in the chemotherapy group (hazard ratio 0.59 [95% CI 0.46–0.76]. Additionally, in the overall population, patients who received nivolumab plus chemotherapy had a PFS of 5.8 months (95% CI, 5.6 to 7.0) compared to 5.6 months (95% CI, 4.3 to 5.9) for patients who received chemotherapy alone [29]. Patients with tumor cell PD-L1 expression of 1% or greater had a significant PFS benefit with nivolumab plus chemotherapy over chemotherapy alone. Treatment-related adverse events of grades 3 or 4 were reported in 47% of patients with nivolumab plus chemotherapy, 32% with nivolumab plus ipilimumab, and 36% with chemotherapy alone.

ATTRACTION-4 is an Asian, randomized, placebo-controlled, phase 2–3 trial that evaluated HER2-negative, unresectable advanced or recurrent gastric or gastroesophageal junction cancer. Patients were randomly assigned to receive chemotherapy with nivolumab or placebo plus chemotherapy. The primary endpoints were PFS and OS. A total of 724 patients were randomly assigned to treatment. The study showed that OS was 17.45 months (95% CI 15.67–20.83) and PFS was 10.45 months (95% CI 8.44–14.75) in the nivolumab plus chemotherapy group, in comparison to OS of 17.15 months (15.18–19.65) and PFS 8.34 months (hazard ratio 0.68 98.51% CI 0.51–0.9, *p* = 0.0007) in the placebo plus chemotherapy arm. The findings suggest that nivolumab combined with chemotherapy improves PFS but not OS [18]. The most common treatment-related grade 3–4 adverse events were decreased neutrophil and platelet counts. 

KEYNOTE-590 is a multicenter, randomized, placebo-controlled, phase 3 trial that evaluated previously untreated, histologically, or cytologically confirmed, locally advanced, unresectable, or metastatic esophageal cancer or Siewert type 1 gastroesophageal junction cancer regardless of PD-L1 status. Patients were randomly assigned to receive pembrolizumab plus chemotherapy or placebo plus chemotherapy. The primary endpoints were OS and PFS in all patients and the subgroup with a PD-L1 CPS of 10 or more. A total of 749 patients were enrolled and underwent randomization. In patients with PD-L1 CPS ≥ 10, the study showed that the median OS and PFS in the pembrolizumab plus chemotherapy were 16.9 and 8.2 months, respectively, in comparison to 11.2 (hazard ratio 0.58) and 4.3 (hazard ratio 0.36) months in the chemotherapy group. The study showed that pembrolizumab plus chemotherapy was superior to placebo plus chemotherapy for OS and PFS patients with esophageal SCC and PD-L1 CPS of 10 or more [19]. However, treatment-related adverse events of grade 3 or higher occurred in 72% of patients in the pembrolizumab plus chemotherapy group versus 68% in the placebo plus chemotherapy group. 

KEYNOTE-062 is a global, randomized, controlled, phase 3 study that evaluates untreated, locally advanced, unresectable, or metastatic gastric or gastroesophageal junction cancer with PD-L1 CPS of 1 or greater. Patients were randomly assigned to receive pembrolizumab or pembrolizumab plus chemotherapy or chemotherapy and placebo. The primary endpoints were PFS and OS. The secondary endpoints included objective response rate (ORR) and duration of response (DOR). A total of 763 patients were selected and underwent randomization. In the CPS ≥ 10 arms, the study showed that OS was 12.3 months (hazard ratio 0.85 *p* = 0.16) and PFS was 5.7 months (hazard ratio 0.73) in the pembrolizumab plus chemotherapy group, in comparison to an OS of 10.8 months and PFS 6.1 months in the chemotherapy group. The study showed that pembrolizumab was non-inferior to chemotherapy, with fewer adverse events observed [30]. The study suggests that pembrolizumab was non-inferior to chemotherapy, and there was no clinically meaningful benefit of pembrolizumab plus chemotherapy vs chemotherapy. 

KEYNOTE-859 is a global, randomized, placebo-controlled, phase 3 trial that evaluates locally advanced or metastatic HER2-negative or gastroesophageal AC with any degree of PD-L1 expression. Patients were randomly assigned to receive pembrolizumab plus chemotherapy or placebo plus chemotherapy. The primary endpoint was OS; secondary endpoints included PFS, ORR, and DOR. A total of 1579 patients were selected and underwent randomization. The median OS was 12.9 months for the pembrolizumab plus chemotherapy group and 11.5 months for the chemotherapy alone group (hazard ratio 0.78, *p* < 0.0001).

Additionally, the median PFS was 6.9 months for the pembrolizumab plus chemotherapy group and 5.6 months for the chemotherapy group (hazard ratio 0.76, *p* < 0.0001). The interim results of the study demonstrated improvement in OS for patients receiving pembrolizumab and chemotherapy [31]. Grade 1 or 2 adverse events, such as immune-related toxicities, especially hypothyroidism, were more common in patients who received pembrolizumab plus chemotherapy (27.1% vs. 9.3%). Grade 3 to 5 treatment-related adverse events occurred in 59.4% of patients in the pembrolizumab arm and 51.1% in the control arm. An interim analysis confirms the OS benefit of first-line immunotherapy plus chemotherapy in advanced gastric cancer. 

ORIENT-16 is a Chinese, randomized, placebo-controlled, phase 3 trial that evaluated advanced gastric and gastroesophageal junction irrespective of PD-L1 expression. Patients were randomly assigned to receive sintilimab, an anti-PD1 monoclonal antibody, or placebo plus chemotherapy. The primary endpoint was OS. A total of 650 patients were selected and underwent randomization. In patients with CPS ≥ 5, the study showed that the median OS in the sintilimab plus chemotherapy group was 18.4 months compared with 12.9 months (hazard ratio 0.66, *p* = 0.0023) for chemotherapy alone. Additionally, patients who received sintilimab plus chemotherapy experienced a PFS of 7.7 months compared to 5.8 months (hazard ratio 0.628, *p* = 0.0002) for patients who received chemotherapy alone. This study demonstrated a superior OS benefit in patients who received sintilimab and chemotherapy and a superior PFS in patients with CPS ≥ 5 [32]. Higher Grade 3 and above adverse events were observed in patients who received sintilimab and chemotherapy. 

Intega is a multicenter, randomized, exploratory phase II clinical trial that evaluated previously untreated HER2-positive esophagogastric AC. Patients were randomly assigned to receive trastuzumab and nivolumab with either ipilimumab or mFOLFOX6. The primary endpoint was to determine the OS advantage of ipilimumab or FOLFOX in combination with nivolumab and trastuzumab. The secondary endpoints were tolerability, PFS and ORR. A total of 88 patients were selected and underwent randomization. In the PD-L1 ≥ 0 arm, the study showed that OS was 12.8 months (hazard ratio 0.91 *p* = 0.89). PFS was 5.1 months (hazard ratio 0.7, *p* = 0.55) in the ipilimumab/trastuzumab/nivolumab group, in comparison to an OS of 11.1 months and PFS of 6.25 months in the FOLFOX/trastuzumab/nivolumab group. The study showed that trastuzumab and nivolumab plus FOLFOX had increased efficacy compared to the trastuzumab regimen, while trastuzumab and nivolumab plus ipilimumab did not improve OS over trastuzumab and chemotherapy [33]. A higher incidence of immune-related adverse events occurred in the ipilimumab group, while a higher incidence of neuropathy was observed in the FOLFOX group. This study suggests that trastuzumab and nivolumab plus FOLFOX have improved efficacy compared to trastuzumab and nivolumab plus ipilimumab. 

Finally, KEYNOTE-811 is a global, randomized, placebo-controlled, phase 3 trial that evaluated previously untreated unresectable, or metastatic HER2-positive gastric or gastroesophageal junction AC. Patients were randomly assigned to receive chemotherapy with pembrolizumab or placebo and trastuzumab. The primary endpoints were PFS and OS. The addition of pembrolizumab to trastuzumab and chemotherapy significantly improved the ORR in HER2-positive gastric cancer [24]. The pembrolizumab group had a 22.7% improvement in ORR compared to the control group. Adverse events were similar between the two groups, with potential immune-mediated adverse events occurring more frequently in the pembrolizumab group. As such, this combination may be a transformative treatment option for HER2-positive gastric or gastroesophageal junction AC.

CheckMate 649, 648, ATTRACTION-4, KEYNOTE-590, KEYNOTE-062, KEYNOTE-859, ORIENT-16, INTEGA, and KEYNOTE-811 were trials that evaluated the efficacy of immunotherapy in combination with chemotherapy or as a monotherapy compared to chemotherapy alone. These studies demonstrate that combinations of immunotherapy and chemotherapy significantly improve OS and PFS in patients with advanced gastric and esophageal cancers (Table 1 below). Adverse events such as immune-related toxicities were generally more common in patients receiving immunotherapy plus chemotherapy but manageable. The combination of CPIs with chemotherapy is considered the standard of care in the first-line setting for treating metastatic gastroesophageal cancers. However, due to shortcomings of predictive biomarkers, there may be a subset of patients who may not derive benefit from the addition of CPIs to chemotherapy.

## 3. Role of Checkpoint Inhibitors in the Second-Line Setting and Beyond

Although there are several systemic therapy options after progression on first-line combination therapies with CPIs, these regimens generally have a modest clinical benefit. Therefore, more effective treatments are necessary. Per the consensus national guidelines, patients with advanced gastroesophageal AC who progress following first-line therapy may be candidates for ramucirumab plus paclitaxel. Other options include FOLFIRI plus ramucirumab or any other chemotherapy backbone which has activity in gastroesophageal cancers. For non-AC malignancies, chemotherapy alone remains the standard of care. Trastuzumab-deruxtecan remains a potential option for second-line therapy in patients with HER2-positive tumors.

Checkpoint inhibitors have a limited role in later lines of therapy once patients progress on upfront treatment. There is not enough data to justify its use beyond the first line. There have been several trials that have investigated CPIs in second-line and beyond for CPI-naïve patients. KEYNOTE-181 was a phase III trial which also investigated second-line treatments for patients with advanced or metastatic esophageal SCC or AC of the esophagus. This trial evaluated the use of pembrolizumab or chemotherapy in these patients. It was observed that the OS was longer for patients receiving pembrolizumab versus chemotherapy with CPS ≥ 10 at 9.3 months versus 6.7 months [34]. There did appear to be some immune-mediated adverse effects and infusion reactions in about 23% of patients in the pembrolizumab group versus 7.4% in the chemotherapy group. This led to the FDA approval of pembrolizumab as a second-line agent for those with recurrent, locally advanced, or metastatic esophageal SCC with tumors expressing PD-L1 [34]. 

In the ATTRACTION-1 trial, nivolumab was evaluated in patients with metastatic or recurrent SCC, esophageal AC, or adenosquamous-cell of the esophagus. This phase II single-arm study used nivolumab as monotherapy (3 mg/kg every two weeks) [35]. There were 65 patients in this study, with a 17% response rate and a median PFS of 1.5 months. The most common adverse effects were increased infection, decreased appetite, increased blood creatinine and dehydration [35]. Overall, ATTRACTION-1 demonstrated that nivolumab has promising efficacy in those with metastatic or recurrent esophageal SCC. 

Another selective IgG4 monoclonal antibody that inhibits PD-1 is camrelizumab. This drug was studied in the randomized, open-label, phase III trial ESCORT, conducted in China with 457 patients diagnosed with esophageal SCC. Patients were randomized to either camrelizumab or chemotherapy. Those who received camrelizumab as monotherapy had an OS of 8.3 months vs an OS of 6.2 months in those who received chemotherapy. Patients who received camrelizumab had some adverse events, such as anemia and liver dysfunction, and seven deaths related to its use [35]. Based on these findings, camrelizumab is now a second-line treatment option for esophageal SCC in China. 

Third-line treatment options for patients with advanced gastric cancers remain limited. The standard of care in this setting currently includes chemotherapy agents. In 2017, pembrolizumab received accelerated approval for treating patients with recurrent, locally advanced, or metastatic gastric or GEJ, although this approval was later rescinded. No CPIs are approved as a third-line agent in refractory gastric cancers. Over the past decade, several clinical trials have evaluated different agents for treating advanced gastric cancer in the third-line setting.

KEYNOTE-059 was a global, phase 2, open-label, single-arm, multicohort study evaluating pembrolizumab monotherapy in patients with previously treated gastric and GEJ cancer who experienced disease progression after two or more lines of therapy. The primary endpoints were ORR and safety. In patients with gastric and GEJ cancer and PD-L1 CPS expression ≥ 1, pembrolizumab monotherapy had a favorable ORR compared to PD-L1-negative patients. The ORR was 15.5% in PD-L1-positive patients, compared to 6.4% in PD-L1-negative patients, and the median OS was 5.8 months versus 4.6 months, respectively. Additionally, the DOR was prolonged in the PD-L1-positive group, with an overall DOR of 8.4 months and a DOR of 16.3 months in PD-L1-positive patients. 46 (17.8%) patients encountered one or more adverse events of grade 3 to 5 attributed to the treatment. Two (0.8%) patients withdrew from the treatment due to treatment-related adverse events, and the study considered two deaths associated with the treatment [36]. These results led to the accelerated approval of pembrolizumab monotherapy as a third-line treatment option for patients with progressive EGC who have PD-L1 CPS expression ≥ 1 in the United States. However, this approval was rescinded in 2021 after post-marketing clinical trials failed to demonstrate significant improvement in OS. [37]. 

ATTRACTION-2 is a multi-national, phase III, randomized, double-blind, placebo-controlled trial assessing nivolumab in patients with advanced gastric or GEJ cancers previously treated with two or more chemotherapy regimens. Four hundred ninety-three patients were randomly assigned to receive either nivolumab monotherapy or placebo. The primary endpoint was OS in the intention-to-treat population. The 12-month OS rates for patients treated with nivolumab and placebo were 26.2% (95% CI 20.7–32.0) and 10.9% (95% CI 6.2–17.0), respectively. In the study, 10% of the 330 patients who received nivolumab experienced grade 3 or 4 treatment-related adverse events, while only 4% of the 161 patients who received placebo had similar adverse events. Of the patients who received nivolumab, 2% died due to treatment-related adverse events, compared to 1% of patients who received a placebo [38]. The results demonstrate that nivolumab may have survival benefits in those with advanced-stage gastric or GEJ cancer, and it has been approved for use in Japan [39]. The study had limitations, such as excluding patients who had received prior treatment with a PD-L1 inhibitor or chemotherapy within four weeks of randomization, which may not represent the general patient population. There was no comparison arm to assess the relative efficacy and safety of nivolumab compared to other treatment options.

Javelin Gastric 300 is a global, phase 3, randomized trial in which 371 patients were randomized to receive either avelumab or chemotherapy (paclitaxel or irinotecan). The study’s primary endpoint was OS. Secondary endpoints were PFS, ORR and safety. The OS was not significantly improved in those receiving avelumab compared to chemotherapy, with an OS of 4.6 months in those receiving avelumab versus 5.0 months in those receiving chemotherapy. Adverse events related to treatment (TRAEs) were reported in 90 patients (48.9%) and 131 patients (74.0%) in the avelumab and chemotherapy groups, respectively. A total of 17 patients (9.2%) in the avelumab group experienced grade ≥3 TRAEs, whereas 56 patients (31.6%) in the chemotherapy group experienced the same [40]. These results suggest that avelumab may be better tolerated by patients than chemotherapy. 

Further studies of immune CPIs in novel combinations are warranted to broaden the options for second and third-line treatment of advanced GEJ cancers.

## 4. Role of Checkpoint Inhibitors in the Perioperative Setting

At diagnosis, approximately 30–40% of patients with gastric cancer are potential candidates for curative surgery [41]. Several clinical trials are exploring the impact of implementing CPI during the perioperative period to improve surgical outcomes.

CheckMate 577 was a randomized, double-blind, placebo-controlled phase 3 trial to evaluate nivolumab as adjuvant therapy in patients with esophageal or gastroesophageal junction cancer with resected (R0) stage II or III esophageal or gastroesophageal junction cancer who had received neoadjuvant chemoradiotherapy and had the residual pathological disease. Patients were randomized in a 2:1 ratio to receive nivolumab or placebo for one year. The primary endpoint was DFS. The median DFS was 22.4 months with nivolumab (95% confidence interval, 16.6 to 34.0), as compared with 11.0 months (95% CI, 8.3 to 14.3) with placebo (hazard ratio for disease recurrence or death, 0.69; 96.4% CI, 0.56 to 0.86; *p* < 0.001). Based on this data, the FDA approved nivolumab for the adjuvant treatment of completely resected esophageal or gastroesophageal junction cancer with residual pathologic disease in patients who have received neoadjuvant chemoradiotherapy. 

In the DANTE phase IIb trial, atezolizumab, an anti-PD-L1 CPI was investigated in patients with resectable localized gastric or gastroesophageal junction AC. Two hundred ninety-five participants were randomized to receive either atezolizumab plus FLOT chemotherapy (5-fluorouracil, folinic acid, oxaliplatin, docetaxel) versus the standard arm of chemotherapy alone. Progression-free survival is still being evaluated. Interim results show the beneficial effects of atezolizumab combined with FLOT against mono-chemotherapy regarding pathologic staging and regression, particularly more pronounced in patients with increased PD-L1 expression [42].

VESTIGE is an open-label phase II clinical trial with 197 participants investigating whether administering an adjuvant treatment with nivolumab plus ipilimumab can enhance DFS in patients diagnosed with stage Ib-IVa gastric and esophagogastric junction AC who have a high-risk of recurrence (defined by ypN1-3 and/or R1 status) after undergoing neoadjuvant chemotherapy and resection [43].

The efficacy of pembrolizumab in the perioperative setting of localized gastric or gastroesophageal junction AC is further being investigated in KEYNOTE-585, a randomized, double-blind, phase III trial with 1007 enrolled participants. This trial compares pembrolizumab plus chemotherapy (cisplatin + capecitabine/5FU) to placebo plus chemotherapy as a neoadjuvant/adjuvant treatment [25]. Notably, the KEYNOTE-585 trial selected the chemotherapy regimen of cisplatin + capecitabine/5FU based off of study results from the FLOT-4 trial, a multi-centre, randomized phase III trial comparing the docetaxel-based therapy with 5-FU, leucovorin, oxaliplatin, and docetaxel (FLOT) against the anthracycline-based triplet therapy of epirubicin, cisplatin, and 5-FU (ECF) as perioperative treatment in patients with resectable gastric or gastroesophageal cancer [44]. Mean OS was 35 months with ECF versus 50 months with FLOT, with improved outcomes noted with FLOT compared to ECF [45].

The MATTERHORN trial is a global, double-blind, phase III trial examining the efficacy of combining durvalumab or placebo therapy with FLOT chemotherapy (fluorouracil + leucovorin + oxaliplatin + docetaxel) in the neoadjuvant and adjuvant setting. There are 958 participants currently enrolled [46].

Avelumab is being investigated in the phase I/II trial (NCT03490292) to study its efficacy as a perioperative treatment in combination with chemotherapy for patients with stage II/III resectable esophageal or gastroesophageal cancer. Phase I of the trial included six patients to evaluate the safety and tolerability of avelumab in combination with chemotherapy (carboplatin, paclitaxel). Phase II of the trial included an expanded cohort of 16 additional participants. The study hypothesis is that co-administration of avelumab and chemotherapy will demonstrate favorable tolerability and contribute to an improved rate of pathological complete response in surgically removed tumor specimens. The trial also hypothesizes that avelumab will decrease the incidence of disease recurrence [47]. Checkpoint inhibitors in the perioperative setting of gastric cancer are presently available exclusively in the context of clinical trials.

## 5. Diagnostic Tests and Biomarkers for Upper Gastrointestinal Cancers

Patients with upper gastrointestinal malignancies that exhibit MSI-H/dMMR have shown improved survival outcomes when treated with CPI in tissue studies [48]. A few diagnostic tests and biomarkers, such as histology, PD-L1 and HER2 expression, can help guide treatment. 

Although SCC is the predominant histological type of esophageal cancer worldwide, in the United States, adenocarcinoma makes up approximately 80% of the cases [49]. The focus on SCC tumors has historically been limited in US studies due to the trial inclusion of exclusive adenocarcinoma [50]. However, in recent years, there has been an increasing representation of squamous histology in clinical trials conducted in the US. For example, the KEYNOTE-590 trial investigated first-line chemotherapy with or without pembrolizumab in advanced esophageal cancer [19], and the CheckMate 648 study exclusively enrolled patients with SCC and demonstrated improved outcomes with PD-1 inhibitor combination as first-line treatment [29]. 

PD-L1 expression is an essential biomarker for upper gastrointestinal malignancies. Several studies have demonstrated improved outcomes in upper gastrointestinal tumors with higher PD-L1 expression, but the benefits have also been observed in all-comer populations [17,19,24,29,30]. It is conceivable that the survival advantage is mainly driven by tumors with increased PD-L1 expression. The CheckMate 649 study showed that the addition of nivolumab to chemotherapy resulted in a significant OS benefit in all patients with esophagogastric AC. However, in the exploratory subgroup analysis, this survival benefit did not achieve statistical significance in tumors with CPS < 5 [17]. While the FDA has approved anti-PD-1 therapy for esophagogastric cancers regardless of PD-L1 expression, the European Commission has restricted the indication to tumors with CPS ≥ 5.

HER2 is another significant biomarker for gastrointestinal malignancies. Approximately 20% of advanced gastric and gastroesophageal junction ACs and 26% of esophageal ACs exhibit HER2 overexpression [24,51]. The KEYNOTE-811 study demonstrated the safe and effective use of anti-HER2 agents in combination with chemo-immunotherapy, leading to FDA accelerated approval for this regimen as a first-line treatment [24]. Interestingly, there may be a synergistic effect between HER2 inhibition and CPIs since trastuzumab has been found to upregulate PD-1 and PD-L1 expression and increase tumour-infiltration lymphocytes [52]. However, there is no available data suggesting a direct correlation between HER2 overexpression and PD-L1 expression [53]. 

Circulating tumor DNA (ctDNA) is an additional marker that can assist physicians in tailoring specific treatments. However, the detection of ctDNA poses challenges due to its limited quantities. In a retrospective cohort study involving 17 individuals diagnosed with stage IIA to IIIB esophageal squamous cell carcinoma, the evaluation of ctDNA was performed both before and following surgical intervention. Only two of the eight patients who exhibited somatic mutations before surgery retained this mutation post-surgery [54]. These findings imply that ctDNA can monitor disease load and minimal residual disease. For upper GI cancers, new-generation sequencing revealed that at least 37% of patients carry somatic mutations (TP53, KRAS) or exhibit gene amplifications, such as HER2, MET, EGFR, and FGFR2 [55,56,57,58]. The potential utilization of ctDNA as a prognostic and predictive indicator for patients with gastric cancer is limited; nonetheless, we anticipate that ongoing prospective observational research (PLAGAST, NCT-02674373) will provide more insight. In the future, integrating ctDNA with imaging may facilitate the assessment of treatment response in patients undergoing therapeutic treatments. 

## 6. Checkpoint Inhibitor Resistance

Checkpoint inhibitor resistance is the phenomenon in which cancer cells become resistant to the effects of CPI. Despite the initial success of CPI treatment in some patients, the cancer cells can adapt and develop mechanisms to evade the immune system’s attack [59]. As a result, the tumors may continue to grow and progress despite ongoing CPI therapy. Several factors and mechanisms described below can contribute to CPI resistance.

Pharmacological blockage of PD-1 or PD-L1 has been the most common mechanism of action of common immunotherapy. The effectiveness of these immunotherapies depends on major factors such as expression in cancer cells, lack of tumor antigens, ineffective antigen presentation, activation of oncogenic pathways, mutations in INF- γ signaling, and factors within the tumor microenvironment including exhausted T cells, Tregs, myeloid-derived suppressor cells, and tumor-associated macrophages [59]. Poorly immunogenic tumors show a lack of response to PD-1/PD-L1 [60]. Loss of antigen-presenting machinery components such as beta-2-microglobulin and HLA is another mechanism to avoid antigen processing and presentation by the tumor [61]. In five cell lines derived from metastatic melanomas with functional loss of beta-2-microglobulin expression, replacement of beta-2-microglobulin was shown to restore antigen processing capabilities of the cells and recognition of tumor by T cells [62]. 

Signaling transduction is another major contributing factor to resistance to immunotherapy. Multiple pathways that contribute to immunosuppression resistance have been reported in the literature, including PI3K/AKT pathway, β-catenin pathway, and JAK/STAT/IFN-γ pathway [63,64,65,66]. The loss of tumor suppressor PTEN enhances PI3K/AKT pathway activation, leading to increased expression of immunosuppressive cytokines that diminish T cell infiltration within tumors [63,67]. This pathway is linked to resistance to CPIs. The WNT pathway, another signaling pathway that stabilizes β-catenin, exhibits defects that result in the activation of the WNT/ β-catenin signaling pathway, which has been linked to increased resistance to PD-1 blockade [64]. The JAK/STAT/IFN-γ pathway is a third signaling pathway that can contribute to CPIs resistance. JAK1/2 regulates the expression of cytokines like CXCL9, CXCL10, and CXCL11 responsible for attracting T cells. In the presence of loss of function mutations in JAK1/2, T-cell infiltration is reduced, and interferon-gamma signaling is lost, ultimately leading to resistance against anti-PD-1 therapy [68]. 

Many patients experience limited or no benefit from CPI, including patients with cancer types that are considered immunogenic. Combination strategies of CPI with chemotherapy, radiotherapy, targeted therapies, or other immunotherapy compounds have been formulated to enhance immune responses and potentially overcome resistance to CPIs. 

## 7. Future Directions

Immune checkpoint inhibitors remain an active and appealing area of research interest in managing gastroesophageal cancers; however, their use is not without potential side effects. CPIs can lead to immune-related adverse events due to their mechanism of action. These adverse events can range from mild to severe, including skin rash, itching, diarrhea, colitis, and fatigue. More severe cases can impact organs such as the liver, lungs, kidneys, and even the nervous and cardiovascular systems [69]. In addition to the potential side effects mentioned earlier, another important consideration is the risk of reactivating autoimmune diseases such as rheumatoid arthritis, lupus, or inflammatory bowel disease [70]. Prompt recognition and management of these side effects are crucial to prevent severe complications and ensure the safety of patients. The management of immune-related adverse events depends on the severity and type of side effects. Healthcare professionals usually grade adverse events using a standardized system, which helps determine the appropriate management approach for each case.

For mild to moderate adverse events, immunosuppressive medication, such as corticosteroids, can help dampen the immune response and alleviate symptoms. The dose and duration of immunosuppression depend on the severity of the adverse event. In severe or life-threatening cases, temporary or permanent discontinuation of CPI therapy may be necessary to ensure patient safety and prevent further complications. Molecular therapies could be developed to selectively target the various genomic subtypes of gastric cancer that have been identified. This can lead to even further promising investigational therapies that will hopefully continue to personalize and broaden our treatment options for this deadly disease. There are several promising CPIs under investigation as monotherapies and in combination with other agents. These trials are summarized in Table 2 below. Furthermore, current guideline treatments are shown in Figure 1 below, helping to guide clinicians in their decision-making process for the management of gastroesophageal and gastric cancers [71].

Despite the recent advances that have been made regarding immuno-oncology, efficacy and survival outcomes generally remain poor, and there is an urgent need for more effective therapies. As our knowledge of tumor characteristics and biology evolves, we can gain valuable insight into novel and personalized treatment strategies that may ultimately improve patient outcomes. Similarly, we are continuing to identify mechanisms of resistance and predictive biomarkers that can further define patients who may or may not respond to such treatments. The role of CPIs in managing locally advanced and metastatic gastroesophageal cancers is continually expanding, and we eagerly await the results of numerous ongoing clinical trials, which may further expand the treatment horizon and improve patient outcomes and quality of life.

## Figures and Tables

**Figure 1 cancers-15-04099-f001:**
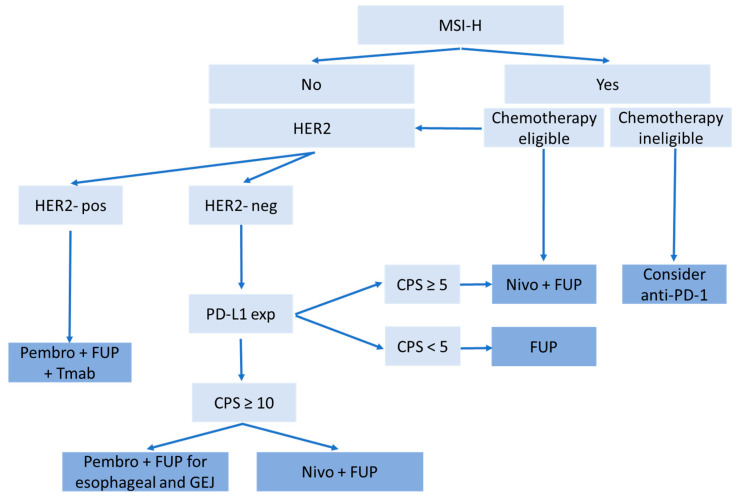
Advanced esophagogastric diagnostic testing and treatment algorithm for advanced esophageal adenocarcinoma, GEJ, or gastric carcinomas.

**Table 1 cancers-15-04099-t001:** Approved Immunotherapies & Major Trials.

Clinical Trial Identifier	Setting	Phase	Site and Histology	Treatment Arm(s)	Primary Endpoint	Patient Selection (Cohort)	Arms (Regimen)	OS			PFS		
								Med	HR	*p*	Med	HR	*p*
CM-649	First line	III	GEJ adenocarcinoma	FP + Nivo vs. FP	PFS and OS (CPS ≥ 5)	CPS ≥ 5 (HER2-neg)	XELOX/FOLFOX + Nivo	14.4	0.71	<0.0001	7.7	0.68	<0.0001
							XELOX/FOLFOX	11.1			6		
						CPS ≥ 1 (HER2-neg)	XELOX/FOLFOX + Nivo	14	0.77	<0.0001	7.5	0.74	-
							XELOX/FOLFOX	11.3			6.9		
						All (HER2-neg)	XELOX/FOLFOX + Nivo	13.8	0.8	<0.0002	7.7	0.77	-
							XELOX/FOLFOX	11.6			6.9		
CM-649	First line	III	GEJ adenocarcinoma	(Nivo/Ipi) vs. FP	OS (CPS ≥ 5)	CPS ≥ 5	Nivo 1 + Ipi 3	11.2	0.89	0.2302	2.8	1.42	-
							XELOX/FOLFOX	11.6			6.3		
						All	Nivo 1 + Ipi 3	11.7	0.91	-	2.8	1.66	-
							XELOX/FOLFOX	11.8			7.1		
CM-648	First line	III	Esophageal squamous cell carcinoma	Nivo + Ipi vs. FP + Nivo vs. FP	PFS and OS (PD-L1 ≥ 1%)	PD-L1 ≥ 1%	Nivo 3 + Ipi 1	20.2	0.46	-	5.4	0.84	-
							FP	9			4.2		
						All	Nivo 3 + Ipi 1	17.6	0.68	-	4.2	1.16	-
							FP	11			4.3		
						PD-L1 ≥ 1%	FP + Nivo	17.3	0.53	-	7	0.56	-
							FP	9			4.2		
						All	FP + Nivo	15.5	0.73	-	6.8	0.76	-
							FP	11			4.3		
Attraction-04	First line	II/III	Gastric or GEJ cancer	Chemo + Nivo vs. Chemo + Placebo	PFS and OS	All (HER2-neg)	XELOX/SOX + Nivo	17.5	0.9	0.257	10.5	0.68	0.0007
							XELOX/SOX	17.2			8.3		
KN-590	First line	III	Esophageal cancer or GEJ squamous cell carcinoma	Pembro + chemo vs. Placebo + chemo	PFS and OS	All	Pembro + chemo	17.6	0.71	-	6.3	0.58	-
							chemo	11.7			6		
						ESCC	Pembro + chemo	17.7	0.69	-	6.4	0.57	-
							chemo	11.7			6.1		
						PD-L1 CPS ≥ 10	Pembro + chemo	16.9	0.58	-	8.2	0.36	-
							chemo	11.2			4.3		
						ESCC PD-L1 CPS ≥ 10	Pembro + chemo	15.8	0.55	-	-	-	-
							chemo	10.9					
KN-062	First line	III	Gastric or GEJ cancer adenocarcinoma	Pembro vs. placebo + FP	PFS (CPS ≥ 1) and OS (CPS ≥ 1)(CPS ≥ 10)	CPS ≥ 1 (HER2-neg)	Pembro + XP/FP	12.5	0.85	0.05	6.9	0.84	0.04
							XP/FP	11.1			6.4		
						CPS ≥ 10 (HER2-neg)	Pembro + XP/FP	12.3	0.85	0.16	5.7	0.73	-
							XP/FP	10.8			6.1		
KN-859	First line	III	Gastric or GEJ cancer adenocarcinoma	(Pembro vs. placebo) + FP cis or FP/CAPOX	PFS and OS	PD-L1 CPS ≥ 1 (HER2-neg)	Pembro + chemo	12.9	0.78	<0.0001	6.9	0.76	<0.0001
							chemo	11.5			5.6		
Orient-16	First line	III	EG adenocarcinoma	Sintilimab vs. placebo + chemo (XELOX)	OS (CPS ≥ 5 and all)	CPS ≥ 5 (HER2-neg)	XELOX + Sint	18.4	0.66	0.0023	7.7	0.63	0.0002
							XELOX	12.9			5.8		
						All (HER2-neg)	XELOX + Sint	15.2	0.77	0.009	7.1	0.64	<0.0001
							XELOX	12.3			5.7		
Intega	First line	II	EG adenocarcinoma	Ipi + Tmab + Nivo vs. FOLFOX + Tmab + Nivo	PFS and OS	HER2 = 0	All	9.65	0.57	0.19	3.7	0.46	0.027
							Ipi + Tmab + Nivo	7.9	0.46	0.16	1.7	0.2	0.0047
							FOLFOX + Tmab + Nivo	32.9	0.83	0.8	9.2	0.79	0.67
						HER2 > 0	All	23.3	0.57	0.19	9.5	0.46	0.027
							Ipi + Tmab + Nivo	23.3	0.46	0.16	8.4	0.2	0.0047
							FOLFOX + Tmab + Nivo	22.1	0.83	0.8	11.2	0.79	0.67
						PD-L1 = 0	All	19.6	1.05	0.92	6.5	1.39	0.48
							Ipi + Tmab + Nivo	11.5	0.91	0.89	3.5	0.7	0.55
							FOLFOX + Tmab + Nivo	32.9	0.98	0.98	13.7	2.71	0.21
						PD-L1 > 0	All	12.8	1.05	0.92	5.15	1.39	0.48
							Ipi + Tmab + Nivo	12.8	0.91	0.89	5.1	0.7	0.55
							FOLFOX + Tmab + Nivo	11.1	0.98	0.98	6.25	2.71	0.21
KN-811	First line	III	Gastric or GEJ adenocarcinoma	(Pembro vs. placebo) + Tmab + FP	PFS and OS	HER2-pos	FP/XELOX + Tmab + Pemb	-	-	-	-	-	-
							FP/XELOX + Tmab	-	-	-	-	-	-

Abbreviations: OS: overall survival; PFS: progression-free survival; med: median (months); HR: hazard ratio; *p*: *p* value; CPS: PD-L1 combined positive score; Pemb: pembrolizumab; Nivo: nivolumab; Sint: sintilimab; Nivo 1: nivolumab 1 mg/kg; Nivo 3: nivolumab 3 mg/kg; ipi 1: ipilimumab 1 mg/kg; ipi 3: ipilimumab 3 mg/kg; GEJ: gastroesophageal junction; EG: esophago-gastric.

**Table 2 cancers-15-04099-t002:** Ongoing Trials involving Immunotherapies.

Clinical Trial Identifier	Line	Phase	Site and Histology	Treatment Arm(s)	Primary Endpoint
IMU-131	Second line	II	GE adenocarcinoma	HER-Vaxx with Ramucirumab + Paclitaxel vs. HER-Vaxx with Pembro	AE and ORR
RECIST	Second line	I	GE adenocarcinoma	Pembro + Lenvatinib	ORR
N-803	Second line	II	GEJ Cancer	Irradiated PD-L1 CAR-NK Cells + Pembrolizumab + N-803	cCR
NCI-2020-05251	First line	I	GEJ Cancer	Pembro + chemoradiation	cCR
TTX-030	First line	I	GEJ Cancer	TTX-030 (Anti-CD39) ± Pembrolizumab ± Budigalimab ± Chemotherapy	AE and DLT
SEQUEL	First line	II	Gastric cancer, GE cancer adenocarcinoma	Pembro + Ramucirumab + Paclitaxel	BORR and PFS
BMS Protocol CA209-76L	First line	II	GE Adenocarcinoma	FOLFOX with nivolumab alone vs. RT with nivolumab	PFS
CA224-060	First line	II	Gastric or GEJ Adenocarcinoma	Relatlimab (Anti-LAG-3) and Nivolumab in Combination With Chemotherapy vs. Nivolumab in Combination W/Chemotherapy	ORR and BOR
AIO-STO-0417	First line	II	Adenocarcinoma of the stomach or GE cancer	modified FOLFOX ± Nivolumab and Ipilimumab vs. FLOT plus Nivolumab.	PFS, ORR, OS
ICONIC	First line	II	oesophagogastric adenocarcinoma	FLOT chemotherapy with the anti-PD-L1 antibody Avelumab, pre-op	pCR rate in surgical specimens
NCI-2018-00946	First line	I	GE Adenocarcinoma	Pembrolizumab	ORR
YO39609	First line	Ib.II	Locally Advanced Unresectable or Metastatic Gastric or Gastroesophageal Junction Cancer or Esophageal Cancer	Multiple Immunotherapy-Based Treatment Combinations	OR and AE
CA209-577	Second line	III	Resected Esophageal, or Gastroesophageal Junction Cancer	Nivolumab vs. Placebo	DFS and OS

Abbreviations: OS: overall survival; PFS: progression-free survival; CRR: clinical response rate; cCR: clinical complete response; DLT: dose-limiting toxicity; BORR: best overall response rate; RT: radiation therapy; pCR: polymerase chain reaction; ORR: objective response rate; BOR: best overall response; OR: objective response; AE: adverse effect; DFS: disease-free survival.

## Data Availability

The data used for the findings of the current study are available on request from the corresponding author.

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
