# Peer review of "Role of Checkpoint Inhibitors in the Management of Gastroesophageal Cancers"

_cancers, 2023, doi:10.3390/cancers15164099_

Round 1

Reviewer 1 Report

The authors provided information and covered various aspects of checkpoint inhibitors role in managing gastroesophageal cancers. It discusses the current treatment options, ongoing clinical trials, diagnostic tests, and biomarkers for these malignancies. The information presented appears to be up-to-date. 

As for novelty, the review summarizes existing knowledge and ongoing research in the field but doesn't introduce new or groundbreaking concepts. It synthesizes information from published studies and clinical trials, which is valuable for clinicians and researchers seeking an overview of the current state of checkpoint inhibitor use in managing gastroesophageal cancers.

Author Response

The authors provided information and covered various aspects of checkpoint inhibitor's role in managing gastroesophageal cancers. It discusses the current treatment options, ongoing clinical trials, diagnostic tests, and biomarkers for these malignancies. The information presented appears to be up-to-date.

As for novelty, the review summarizes existing knowledge and ongoing research in the field but doesn't introduce new or groundbreaking concepts. It synthesizes information from published studies and clinical trials, which is valuable for clinicians and researchers seeking an overview of the current state of checkpoint inhibitor use in managing gastroesophageal cancers.

Thank you for your positive review of our article, which covers various aspects of the checkpoint inhibitor's role in managing gastroesophageal cancers. While we primarily focused on summarizing existing knowledge and ongoing research in the field, we acknowledge the feedback regarding the need to introduce more novel concepts. Our article aims to provide a comprehensive overview of different trials and briefly review upcoming trials. We highly appreciate your valuable feedback, which will help us improve our work in the future.

Reviewer 2 Report

In this case review Karim et al, have written a thorough and comprehensive review of the role of checkpoint inhibitors (CPIs) in the setting of gastroesophageal cancers. The manuscript covers a wide spectrum of research including various clinical trials studying the effectiveness of different CPIs.

Below are some of my suggestions to help readers understand better the message of this case review which is otherwise well-written:

1.    The authors might consider providing a brief background of the mechanism of action for CPI, further enhancing the article’s educational value.

2.    Also, it would be great if they could add a figure to support point 1.

3.    In the future direction, the addition of potential side effects of CPI treatment and management would add to the completeness of the review.

With the proposed amendments, I believe this review would be a valuable addition to the literature.

Author Response

In this case review Karim et al, have written a thorough and comprehensive review of the role of checkpoint inhibitors (CPIs) in the setting of gastroesophageal cancers. The manuscript covers a wide spectrum of research including various clinical trials studying the effectiveness of different CPIs.

Below are some of my suggestions to help readers understand better the message of this case review which is otherwise well-written:

  1. The authors might consider providing a brief background of the mechanism of action for CPI, further enhancing the article’s educational value.

The mechanism of action for CPIs involves blocking specific checkpoint molecules, such as PD-1 or CTLA—4. When these checkpoints are blocked, the immune system can recognize and target cancer cells more effectively, leading to tumor regression.

  1. Also, it would be great if they could add a figure to support point 1.

Could you please provide more specific details on what content the figure should include? This will help me better understand your expectations and create a figure that aligns with your preferences

  1. In the future direction, the addition of potential side effects of CPI treatment and management would add to the completeness of the review.

Added the following under section 7:

Immune checkpoint inhibitors remain an active and appealing area of research interest in the management of gastroesophageal cancers; however, their use is not without potential side effects. CPIs can lead to immune-related adverse events due to their mechanism of action. These adverse events can range from mild to severe and may include skin rash, itching, diarrhea, colitis, and fatigue. In more severe cases, it can impact organs such as the liver, lungs, kidneys, and even the nervous and cardiovascular systems [72]. In addition to the potential side effects mentioned earlier, another important consideration is the risk of reactivating autoimmune disease such as rheumatoid arthritis, lupus, or inflammatory bowel disease [73]. Prompt recognition and management of these side effects are crucial to prevent severe complications and ensure the safety of patients. The management of immune related adverse events depends on severity and type of side effect. Healthcare professionals usually grade adverse events using a standardized system, which helps determine the appropriate management approach for each case. For mild to moderate adverse events, the use of immunosuppressive medication, such as corticosteroids, can help dampen the immune response and alleviate symptoms. The dose and duration of immunosuppression depend on the severity of the adverse event. In severe or life-threatening cases, temporary or permanent discontinuation of CPI therapy may be necessary to ensure patient safety and prevent further complications. Molecular therapies could be developed to selectively target the various genomic subtypes of gastric cancer that have been identified, which can lead to even further promising investigational therapies that will hopefully continue to personalize and broaden our treatment options for this deadly disease. There are several promising CPIs under investigation as monotherapies and in combination with other agents. These trials are summarized in table 2 below. Furthermore, current guideline treatments are shown in Figure 1 below, helping to guide clinicians in their decision-making process for the management of gastroesophageal and gastric cancers [74].

Reviewer 3 Report

Very interesting and well written review. Maybe some figures could improve the quality of the paper. 

The authors should comment deeply on the safety profile of these drugs, with particular reference to the risk of reactivation of pre-existing autoimmune disease (cite the recent MA: PMID: 33314269)

Did the authors check trialgov for ongoing RCTs in the field?

Author Response

Very interesting and well written review. Maybe some figures could improve the quality of the paper.

The authors should comment deeply on the safety profile of these drugs, with particular reference to the risk of reactivation of pre-existing autoimmune disease (cite the recent MA: PMID: 33314269)

Added the following under section 7:

Immune checkpoint inhibitors remain an active and appealing area of research interest in the management of gastroesophageal cancers; however, their use is not without potential side effects. CPIs can lead to immune-related adverse events due to their mechanism of action. These adverse events can range from mild to severe and may include skin rash, itching, diarrhea, colitis, and fatigue. In more severe cases, it can impact organs such as the liver, lungs, kidneys, and even the nervous and cardiovascular systems [72]. In addition to the potential side effects mentioned earlier, another important consideration is the risk of reactivating autoimmune disease such as rheumatoid arthritis, lupus, or inflammatory bowel disease [73]. Prompt recognition and management of these side effects are crucial to prevent severe complications and ensure the safety of patients. The management of immune related adverse events depends on severity and type of side effect. Healthcare professionals usually grade adverse events using a standardized system, which helps determine the appropriate management approach for each case. For mild to moderate adverse events, the use of immunosuppressive medication, such as corticosteroids, can help dampen the immune response and alleviate symptoms. The dose and duration of immunosuppression depend on the severity of the adverse event. In severe or life-threatening cases, temporary or permanent discontinuation of CPI therapy may be necessary to ensure patient safety and prevent further complications. Molecular therapies could be developed to selectively target the various genomic subtypes of gastric cancer that have been identified, which can lead to even further promising investigational therapies that will hopefully continue to personalize and broaden our treatment options for this deadly disease. There are several promising CPIs under investigation as monotherapies and in combination with other agents. These trials are summarized in table 2 below. Furthermore, current guideline treatments are shown in Figure 1 below, helping to guide clinicians in their decision-making process for the management of gastroesophageal and gastric cancers [74].

Did the authors check trialgov for ongoing RCTs in the field?

Yes we mentioned many ongoing trials that were summarized in table 2

Reviewer 4 Report

The review by Karim et al.  summarizes the clinical trials where checkpoint inhibitors are used for treatment of Gastroesophageal Cancers.

The review is well-researched, well-written with adequate examples of clinical trials. I specifically like how the introduction started with esophageal cancer and further describe how checkpoint inhibitors are used in first line/second line and preoperative setting with relevant examples from the clinical trials.

Some very minor comments are as follows:

1.     A separate section on mechanism of checkpoint inhibitor resistance would add more value to the article.

2.     Also a small paragraph on how to overcome checkpoint inhibitor resistance would be good to discuss. The authors can also write about what approaches and advancements are made and can be made to overcome the resistance.

3.      What do the authors think of circulating tumor DNA (CtDNA) as a useful biomarker in monitoring disease response/recurrence?

Author Response

The review by Karim et al.  summarizes the clinical trials where checkpoint inhibitors are used for treatment of Gastroesophageal Cancers.

The review is well-researched, well-written with adequate examples of clinical trials. I specifically like how the introduction started with esophageal cancer and further describe how checkpoint inhibitors are used in first line/second line and preoperative setting with relevant examples from the clinical trials.

Some very minor comments are as follows:

  1. A separate section on mechanism of checkpoint inhibitor resistance would add more value to the article.

Added the following under section 6:

Checkpoint inhibitor resistance is the phenomenon in which cancer cells become resistant to the effects of CPI. Despite the initial success of CPI treatment in some patients, the cancer cells can adapt and develop mechanisms to evade the immune system’s attack [62]. As a result, the tumors may continue to grow and progress despite ongoing CPI therapy. Several factors and mechanisms described below can contribute to CPI resistance.

Pharmacological blockage of PD-1 or PD-L1 has been the most common mechanism of action of common immunotherapy. The effectiveness of these immunotherapies depends on major factors such as expression in cancer cells, lack of tumor antigens, ineffective antigen presentation, activation of oncogenic pathways, mutations in INF- γ signaling, and factors within the tumor microenvironment including exhausted T cells, Tregs, myeloid derived suppressor cells, and tumor-associated macrophages [62]. Poorly immunogenic tumors are known to show a lack response to PD-1/PD-L1 [63]. Loss of antigen-presenting machinery components such as beta-2-microglobulin and HLA is another mechanism to avoid antigen processing and presentation by tumor [64]. In five cell lines derived from metastatic melanomas with functional loss of beta-2-microglobulin expression, replacement of beta-2-microglobulin was shown to restore antigen processing capabilities of the cells, as well as recognition of tumor by T cells [65].

Signaling transduction is another major contributing factor to resistance to immunotherapy. Multiple pathways that contribute to immunosuppression resistance have been reported in the literature, including PI3K/AKT pathway, β-catenin pathway, and JAK/STAT/IFN-γ pathway [66,67,68,69]. The loss of tumor suppressor PTEN enhances PI3K/AKT pathway activation, leading to increased expression of immunosuppressive cytokines that diminish T cell infiltration within tumors [66,70]. This pathway is linked to resistance to CPIs. The WNT pathway, another signaling pathway that stabilizes β-catenin, exhibits defects that result in the activation of the WNT/ β-catenin signaling pathway, which has been linked to increased resistance to PD-1 blockade [67]. The JAK/STAT/IFN-γ pathway, a third signaling pathway that can contribute to CPIs resistance. JAK1/2 regulates the expression of cytokines like CXCL9, CXCL10, and CXCL11 responsible for attracting T cells. In the presence of loss of function mutations in JAK1/2, T-cell infiltration is reduced, interferon gamma signaling is lost, ultimately leading to resistance against anti-PD-1 therapy [71].

  1. Also a small paragraph on how to overcome checkpoint inhibitor resistance would be good to discuss. The authors can also write about what approaches and advancements are made and can be made to overcome the resistance.

Added the following under section 6:

Many patients experience limited or no benefit with CPI, including patients with cancer types that are considered immunogenic. Combination strategies of CPI with chemotherapy, radiotherapy, targeted therapies or other immunotherapy compounds have been formulated to enhance immune responses and potentially overcome resistance to CPIs.

  1. What do the authors think of circulating tumor DNA (CtDNA) as a useful biomarker in monitoring disease response/recurrence?

Added the following at the end of section 5:

Circulating tumor DNA (ctDNA) serves as an additional marker that can assist physicians in tailoring specific treatments, however the detection of ctDNA poses challenges due to its limited quantities. In a retrospective cohort study involving 17 individuals diagnosed with stage IIA to IIIB esophageal squamous cell carcinoma, the evaluation of ctDNA was performed both prior to and following surgical intervention. Of the eight patients who exhibited somatic mutations prior to surgery, only two retained this mutation post-surgery [57]. These findings imply that ctDNA can be used to monitor disease load and minimal residual disease. For upper GI cancers, new generation sequencing revealed that at least 37% of patients carry somatic mutations (TP53, KRAS) or exhibit gene amplifications, such as HER2, MET, EGFR, and FGFR2 [58,59,60,61]. The potential utilization of ctDNA as a prognostic and predictive indicator for patients with gastric cancer is limited; nonetheless, we anticipate that ongoing prospective observational research (PLAGAST, NCT-02674373) will provide more insight. In the future, the integration of ctDNA with imaging may facilitate the assessment of treatment response in patients undergoing therapeutic treatments.

Round 2

Reviewer 3 Report

The revised version of the manuscript is OK. Thank you!